# Design and Preclinical Validation of an Anti-B7-H3-Specific Radiotracer: A Non-Invasive Imaging Tool to Guide B7-H3-Targeted Therapies

**DOI:** 10.3390/ph18101477

**Published:** 2025-09-30

**Authors:** Cyprine Neba Funeh, Fien Meeus, Niels Van Winnendael, Timo W. M. De Groof, Matthias D’Huyvetter, Nick Devoogdt

**Affiliations:** 1Molecular Imaging and Therapy (MITH) Laboratory, Vrije Universiteit Brussel, 1090 Brussels, Belgium; 2Laboratory for Molecular and Cellular Therapy (LMCT), Translational Oncology Research Center (TORC), Vrije Universiteit Brussel, 1090 Brussels, Belgium

**Keywords:** B7-H3 protein, single domain antibodies, SPECT, cancer, targeted therapy, nuclear imaging

## Abstract

**Background:** B7-H3, an immunoregulatory protein of the B7 family, has been associated with both anti-cancer immunity and tumor promotion, with its expression commonly correlated with poor prognosis. Although it is frequently expressed across cancers, its heterogeneity may limit the effectiveness of B7-H3-targeted therapies. Consequently, a sensitive and non-invasive method is needed to assess B7-H3 expression for patient selection and stratification. Single-domain antibody fragments (sdAbs) offer a promising platform for developing such a diagnostic tool. **Methods:** To generate B7-H3 sdAbs, two Ilamas were immunized with the recombinant human B7-H3 protein. Positive clones were selected through Phage biopanning and characterized for thermal stability, binding specificity, and affinity to human and murine B7-H3 proteins. Selected sdAbs were radiolabeled with Technetium-99m (^99m^Tc) and evaluated for B7-H3 detection in two xenograft tumor models using micro-SPECT/CT imaging and dissection studies. **Results:** Sixteen purified sdAbs bound specifically to recombinant B7-H3 proteins and cells expressing native B7-H3 antigens, with nanomolar affinities. The four best-performing sdAbs bound promiscuously to tested mouse and human B7-H3 isoforms. Lead sdAb C51 labeled with ^99m^Tc displayed specific accumulation across two human B7-H3^+^ tumor models, achieving high contrast with a tumor-to-blood ratio of up to 10 ± 3.16, and a tumor uptake of up to 4.96 ± 1.4%IA/g at 1.5 h post injection. **Conclusions:** The lead sdAb enabled rapid, specific, and non-invasive imaging of human B7-H3^+^ tumors. Its isoform promiscuity supports broad applicability across cancers expressing different human B7-H3 isoforms. These results support further development for clinical translation to enable patient selection and improved B7-H3-targeted therapies.

## 1. Introduction

Targeted therapies have revolutionized the landscape of cancer management, providing patients with improved overall survival and quality of life. Targeted therapies exploit specific molecular patterns/markers within cancer cells to deliver cytotoxic drugs or other modalities that can kill or block the growth and spread of cancer cells [1,2]. This paradigm shift towards a personalized approach enables the selective treatment of patients with tumors that express a molecular target of interest, thereby optimizing the therapeutic efficacy while minimizing off-target effects. Therefore, identifying an ideal target is crucial for developing and deploying targeted therapies [3]. B7 homolog 3 (B7-H3 or CD276) has emerged as an interesting target, exhibiting high expression across a spectrum of solid and hematological malignancies, including melanoma, ovarian, prostate, lung, pancreatic, breast, colorectal, and glioblastoma (GBM), with low to no expression in normal healthy tissues [4]. Moreover, B7-H3 is expressed in primary, metastatic [5], and recurrent tumors [6], as well as in the tumor stroma, vasculature, and cancer-initiating cells [7].

B7-H3 is a type 1 transmembrane glycoprotein and exists in two different isoforms in humans: a short isoform (h2IgB7-H3) with one variable and constant (VC) domain pair, and a predominant long isoform (h4IgB7-H3), with two VC pairs due to exon duplication (Appendix A) [8]. Mice express the 2IgB7-H3 variant (m2IgB7-H3), having close sequence similarity to the human B7-H3 proteins. These proteins are highly conserved, with their extracellular domains exhibiting a 91–94% amino acid identity across species (Appendix A), highlighting their evolutionary role in immune regulation. B7-H3 contributes to tumor progression through immune and non-immune mechanisms [9]. Although its receptor remains unidentified, B7-H3 can inhibit T-cell activation [10], promote tumor proliferation, invasion, and migration through the ERK1/2, PI3K-AKT, NF-kB, and the JAK2/STAT3 signaling pathways [9]. It also supports angiogenesis and chemoresistance, correlating with an overall poor prognosis for cancer patients [4]. Given the significant role B7-H3 plays in the prognosis of cancer, it has emerged as an attractive target for targeted therapies.

Several B7-H3-targeted therapies are under development at preclinical and clinical levels [11]. They include monoclonal antibodies (mAbs) (NCT01391143, NCT02982941), antibody–drug conjugates (ADCs) (NCT05276609) [12], CAR-T cells (NCT06482905) [13], and targeted radionuclide therapies (TRTs) [14]. Despite encouraging outcomes, significant improvements are required to enhance the efficacy and overall effectiveness of these therapies. One of the main challenges is the variability in B7-H3 expression observed in patients, which may influence therapy response. Michelakos et al. pooled 94 studies reporting B7-H3 expression frequencies, assessed by immunohistochemistry (IHC) in a total of 26,703 patients across 21 malignancies, and found a B7-H3 expression frequency of 33.0% to 91.8%, with a cumulative B7-H3 positivity rate of 59.5% (15,877/26,703 patients) [15]. Moreover, B7-H3 expression exhibits significant heterogeneity both intratumorally and between patients with IHC staining variations ranging from negative, weak, moderate, and strong across various cancers [5,16,17,18]. Therefore, advancing the clinical impact of B7-H3-directed therapies requires the development of a sensitive diagnostic method for the appropriate selection and stratification of patient subgroups who can benefit from B7-H3 therapies.

Biopsies for IHC are the standard diagnostic method to assess B7-H3 expression. Unfortunately, IHC is unreliable in evaluating the expression of B7-H3 due to the differences in the target expression between primary and metastatic cells [19,20]. Moreover, biopsies are invasive procedures for patients, and sometimes, tumors are not accessible for a clean biopsy. This is further exacerbated by the limitations of IHC in providing a comprehensive mapping of metastatic lesions in a patient. In contrast, non-invasive nuclear imaging can offer an extensive and robust alternative approach for patient selection and stratification, circumventing the limitations of IHC. A few preclinical studies have demonstrated the utility of non-invasive imaging of B7-H3. For example, zirconium-89 (^89^Zr)-labeled anti-B7-H3 antibodies [21,22] and Technetium-99m (^99m^Tc)-labeled affibodies [23,24] have been used as PET and SPECT tracers for imaging B7-H3^+^ tumors. These preclinical results demonstrated the great potential of these tools as non-invasive methods for assessing the expression of B7-H3 in malignant lesions. However, the large sizes of mAbs (~150 kDa) contribute to prolonged blood circulation, resulting in high blood and non-target organ signals. This gives poor tumor-to-background ratios at early time points, limiting the application of mAbs for same-day imaging. Moreover, the ability of the tracers to recognize both human B7-H3 isoforms was not reported, which may limit their sensitivity and clinical applicability for detecting tumors that express the h2IgB7-H3 isoform. Therefore, an unmet need persists for a time-efficient, non-invasive nuclear imaging strategy capable of detecting tumors expressing both human B7-H3 isoforms.

Alternate vectors with small sizes that favor rapid tumor accumulation, rapid blood clearance, and high contrast imaging at early time points have been investigated for nuclear imaging. These include antibody fragments, scaffold proteins, peptides, small molecules, nanoparticles, and sdAbs [25]. sdAbs, derived from camelid heavy-chain-only antibodies, provide an attractive alternative with ideal properties compared to mAbs. sdAbs are small (12–15 kDa), exhibit rapid pharmacokinetics, penetrate deeply and uniformly into tumors, and are rapidly cleared from the body through the kidneys. This results in rapid imaging, low radiation burden, and high contrast images [26]. For example, anti-HER2 sdAbs have been evaluated as PET and SPECT tracers for detecting primary and metastatic breast cancer lesions in phase I and II clinical trials [27,28].

Here, we describe the development and preclinical validation of cross-reactive anti-B7-H3 sdAbs. We validated their specific binding to recombinant B7-H3 protein isoforms and in various cancer cell lines that endogenously express B7-H3 protein. Additionally, the binding affinities and stabilities of the sdAbs were evaluated. We further validated their diagnostic potential for nuclear imaging by radiolabeling with ^99m^Tc for SPECT imaging in B7-H3-expressing melanoma and GBM xenograft tumor models.

## 2. Results

### 2.1. Generation and Selection of Anti-B7-H3 sdAbs

Anti-B7-H3 sdAbs were generated by immunizing two Ilamas with the recombinant extracellular domain of the human long B7-H3 isoform (h4IgB7-H3 protein), producing two independent libraries, one from each animal. The aim was to identify sdAbs that can recognize all B7-H3 isoforms expressed in mice and humans. The amino acid sequences of the two human and murine B7-H3 proteins are highly conserved (Appendix A); therefore, sdAbs generated against the h4IgB7-H3 would have a high prospect of recognizing all isoforms. Two generations of biopanning experiments were conducted using varying stringency conditions to enhance the selection of high-affinity binders.

For the first generation of biopanning, an in-solution panning protocol was used with 3 panning rounds for each library. An antigen concentration of 100 nM was used for the first and second panning rounds, and 10 nM for the third. A 10-cycle washing procedure was applied in all rounds, with a 5 min interval between each cycle for rounds 1 and 2, and a 10 min interval for round 3. ELISA screening resulted in 81 unique sdAbs, grouped into 12 families. Crude periplasmic extracts (PE) of the 81 sdAbs were used to screen their binding to the 3 B7-H3 recombinant proteins (h4IgB7-H3, h2IgB7-H3, m2IgB7-H3) by ELISA, with 63% of the sdAbs revealing cross-reactivity to all the proteins (Appendix A). Flow cytometry binding analysis of the 81 sdAbs to human LN-229 cells (confirmed to express B7-H3, Figure 1A) demonstrated binding of 50 sdAbs to membrane B7-H3 (Appendix A). Based on these results, the 13 best-performing sdAbs belonging to 6 families were selected, produced, and purified with a C-terminal hexa-histidine (6-His) tag. An anti-B7-H3 sdAb (Nb0), described in a patent [29] and recently characterized for NanoCAR-T therapy [13], was used as a benchmark for sdAb screening. Meanwhile, an irrelevant sdAb (R3B23), binding to the 5T2 multiple myeloma M-protein [30] was used as a negative control. The cross-reactivity of the 13 purified sdAbs to the various recombinant proteins was first confirmed by ELISA (Appendix A), followed by an assessment of cell binding affinity using flow cytometry on human U87-MG and SK-OV-3 cells (Appendix A), previously confirmed to express B7-H3 (Figure 1B, C). Flow cytometry analysis also revealed the specific binding of Nb0 and no binding of the irrelevant control sdAb R3B23. Since radiochemistry protocols often require incubation at elevated temperatures, the thermal stability of the sdAbs was assessed (Appendix A and Table 1). The sdAbs exhibited thermal stability for radiolabeling applications, with melting temperatures ranging from 52.0 °C to 78.0 °C. Subsequently, the affinity of the sdAbs were evaluated on recombinant h4IgB7-H3 protein by surface plasmon resonance (SPR). The binding affinities of the 13 sdAbs ranged from 67 nM to 341 nM (Table 1), with Nb0 performing better, recording a K_D_ value of 14 nM. Although all sdAbs exhibited favorable association rates (k_a_ in the 10^5^–10^6^ 1/M.s range), their dissociation rates (k_d_ ranging from 0.01 to 0.1 1/s) were modest, resulting in mediocre binding affinities. Even though Nb0 had a better k_D_, it displayed a similar interaction profile to our sdAb panel (Appendix A). Two sdAbs, Nb44 and Nb55, were chosen as the best-performing sdAbs from this first-generation biopanning for further characterization. Due to the suboptimal binding affinities obtained, second-generation biopanning was conducted with more stringent conditions to select new sdAbs with better affinities.

For the second-generation biopanning, 4 rounds of in-solution panning were employed. 100 nM of protein was used for the 1st panning round, 10 nM for the 2nd, followed by 1 nM and 0.1 nM for rounds 3 and 4, respectively. For rounds 1 to 3, a 10-cycle washing protocol was used, with the interval between washes progressively increased from 5, 10, and 30 min, respectively. For round 4 panning, a 5-cycle washing protocol was employed with a 60 min interval between each cycle, ending with an overnight washing step. 1 mL washing buffer was used for rounds 1 and 2 and then increased progressively to 2 mL and 5 mL for rounds 3 and 4, respectively. The second-generation panning experiment identified 37 unique sdAbs, grouped into six families based on CDR3 similarities. Four of the six families were homologous to the first-generation binders, and two new families were identified. Based on PE ELISA with the 3 B7-H3 recombinant proteins (Appendix A) and binding to B7-H3^+^ human U87-MG cells (Appendix A), the 3 best sdAbs belonging to 2 different families were selected for further characterization. After production and purification with a C-terminal 6-His tag, a binding ELISA confirmed the 3 sdAbs as cross-reactive to murine and human B7-H3 proteins (Appendix A). The specific binding of B7-H3 on U87-MG and SK-OV-3 cells was confirmed (Appendix A), and their thermal stability was assessed in a thermofluor assay (Appendix A). The affinity assessment on recombinant h4IgB7-H3 protein revealed improved binding affinities (K_D_ values of 21 nM to 103 nM) compared to the first-generation sdAbs (Table 1, Appendix A). From this second-generation biopanning, sdAbs C51 and C80 were chosen as the best binders for subsequent characterization.

### 2.2. Lead Anti-B7-H3 sdAbs Are Cross-Reactive and Efficiently Radiolabeled with ^99m^Tc

The four best performing sdAbs from both generations (Nb44, Nb55, C51, C80) alongside Nb0 and R3B23 were taken for further evaluation. First, their binding to the 3 B7-H3 recombinant proteins was assessed by SPR. Results revealed the cross-reactivity of the 4 sdAbs on all the B7-H3 isoforms, supporting previous ELISA results, with fast k_a_ (association rates) and fast k_d_ (dissociation rates) (Figure 2, Appendix A). Notably, Nb0 showed poor or no binding to the h2IgB7-H3 isoform but exhibited higher affinity for the m2IgB7-H3 isoform with a measured affinity of 9.0 ± 0.7 nM. Additionally, the ability of the sdAbs to target membrane-bound m2IgB7-H3 was assessed using the CT26 murine cell line, which was lentivirally transduced to overexpress m2IgB7-H3 [13]. First, we confirmed the expression of m2IgB7-H3 using a commercially available anti-mouse B7-H3 antibody by flow cytometry (Figure 3A). Figure 3B revealed that all sdAbs specifically bound to the m2IgB7-H3 membrane protein with significantly higher delta mean fluorescence intensities (ΔMFIs) compared to R3B23 (Figure 3C). Of note, Nb0 demonstrated significantly higher ΔMFI compared to Nb44, Nb55, C51, and C80. In contrast, R3B23 showed no binding, further supporting the specificity of the sdAbs.

Next, the sdAbs alongside Nb0 and R3B23 were developed into radiotracers by site-specifically radiolabeling them with Technetium-99m (^99m^Tc) on their C-terminal 6-His tags. Calculated decay-corrected radiochemical yields (RCYs) were >60%, and the resulting radiochemical purities (RCPs), as assessed by iTLC, were >95% (Appendix A). The ability of the radiotracers ([^99m^Tc]Tc-sdAb) to specifically interact with human B7-H3 was assessed in a radioligand binding assay using U87-MG and SK-OV-3 cells. To this end, cells were incubated with 100 nM [^99m^Tc]Tc-sdAb, with or without a 100-fold molar excess of unlabeled sdAb to assess non-specific interaction. Figure 4 shows that all radiotracers specifically bound B7-H3 on both cell lines, although quantitative differences among sdAbs and cell lines can be noted. SK-OV-3 cells have been shown to express both h4IgB7-H3 and h2IgB7-H3 isoforms, while U87-MG primarily expresses h4IgB7-H3 [31]. The co-expression of both isoforms in SK-OV-3 may result in a dynamic, favorable binding to the SK-OV-3 cells.

### 2.3. [^99m^Tc]Tc-sdAbs Allow Specific Non-Invasive Imaging of B7-H3 Expressing Tumors

To evaluate the capability of the sdAbs for non-invasive imaging of B7-H3^+^ tumors, the four lead sdAbs, alongside Nb0 and R3B23, were radiolabeled with ^99m^Tc, and their ability to accumulate in B7-H3 expressing tumors was assessed in a subcutaneous GBM tumor model. Initially, we evaluated the in vivo expression of B7-H3 in implanted U87-MG tumors using single-cell suspensions by flow cytometry analysis. Appendix A shows consistent B7-H3 expression across small to large tumor volumes, validating the suitability of the U87-MG tumor model for in vivo characterization of the sdAbs.

Next, nude mice bearing subcutaneous U87-MG tumors on the flanks were intravenously (i.v.) injected with ~5 µg (54 ± 12.7 MBq) of [^99m^Tc]Tc-sdAbs. 1 h p.i., micro-SPECT/CT imaging was performed, followed by ex vivo biodistribution analysis at 1.5 h p.i., achieved by harvesting the tumors and different organs/tissues and subsequent quantification of the radioactivity using a gamma counter.

Micro-SPECT/CT revealed variable signals in the tumor for all [^99m^Tc]Tc-sdAbs, including the [^99m^Tc]Tc-Nb0 compound, compared to negligible tumor uptake for [^99m^Tc]Tc-R3B23 (Figure 5A). Results also displayed high signals in the kidneys and bladder for all compounds, as explained by their blood clearance through the renal route. In contrast, all compounds demonstrated low signals in all normal organs and tissues, except for [^99m^Tc]Tc-C80, with elevated signals in the liver. Ex vivo biodistribution results aligned with micro-SPECT/CT images demonstrate significantly higher tumor uptake for all [^99m^Tc]Tc-sdAbs (except for [^99m^Tc]Tc-C80), when compared to the [^99m^Tc]Tc-R3B23 (Figure 5B). Additionally, we observed low ex vivo uptake in normal organs and tissues (<1% IA/g) (Figure 5C and Appendix A). Notably, [^99m^Tc]Tc-C51 displayed the highest tumor uptake of 4.96 ± 1.4%IA/g, better than the benchmark sdAb tracer [^99m^Tc]Tc-Nb0 (3.79 ± 0.8%IA/g).

Next, the tumor-to-blood, tumor-to-background, and tumor-to-brain ratios for all the sdAbs were assessed and compared to those of the irrelevant sdAb. [^99m^Tc]Tc-C51 displayed better uptake ratios across all tested conditions, compared to the benchmark compound [^99m^Tc]Tc-Nb0. Based on in vitro and in vivo data, C51 performed better than all compounds and was selected as the ultimate lead candidate for imaging (Figure 6).

To demonstrate the versatility of our lead sdAb (C51) and its general relevance for non-invasive imaging of B7-H3 tumors, sdAb C51, alongside the irrelevant control R3B23, were radiolabeled with ^99m^Tc and their tumor targeting potential evaluated in a melanoma xenograft model. Nude mice (n = 3) bearing 624MEL subcutaneous tumors (B7-H3 positive; Figure 7A) were intravenously injected with ~5 µg of [^99m^Tc]Tc-C51 (53.5 ± 2.0 MBq) or [^99m^Tc]Tc-R3B23 (57.5 ± 2.7 MBq). One-hour p.i. micro-SPECT/CT imaging was performed, followed by ex vivo dissection studies at 1.5 h p.i.

SPECT/CT images (Figure 7B) revealed no accumulation in the tumor for the [^99m^Tc]Tc-R3B23 tracer, in contrast to specific accumulation for [^99m^Tc]Tc-C51. Both tracers also reveal no signals in non-target organs except for the kidneys and bladder due to the rapid clearance of the tracers. Ex vivo dissection studies (Figure 7D) displayed significantly higher tumor uptake (1.44 ± 0.11%IA/g) for the [^99m^Tc]Tc-C51 compared to [^99m^Tc]Tc-R3B23 (0.14 ± 0.03%IA/g). In addition, ex vivo uptake in normal organs and tissues (Figure 7C, Appendix A) remained low (<0.5%IA/g) except for the kidneys. Tumor-to-blood (T/B) and tumor-to-muscle (T/M) ratios (Figure 7E) for the [^99m^Tc]Tc-C51 tracer displayed significantly higher values compared to the irrelevant control.

## 3. Discussion

Identifying suitable patients before or early during targeted therapies is crucial for maximizing clinical outcomes for patients and enhancing treatment effectiveness. Nuclear imaging has demonstrated clinical utility in providing a comprehensive assessment of the spatiotemporal expression of a molecular target of interest in cancer lesions, a valuable requirement in deploying targeted therapies. This capability supports effective patient selection and stratification, enabling clinicians to make the right intervention decisions.

To this end, the growing interest in novel therapies directed against the B7-H3 protein requires a reliable diagnostic method that would allow clinicians to monitor B7-H3 expression levels effectively. Moreover, although B7-H3 represents a promising pan-cancer target, with encouraging safety data from clinical trials, its variable and heterogeneous expression across cancer types remains a major hurdle and should be carefully considered for effective therapies. For example, Zhang et al. performed IHC staining on 209 human tumor samples representing 17 different cancer types. Despite reporting a cumulative B7-H3 expression of 66%, 34% of the samples had negative scores. Among the positive samples, 18% exhibited strong staining, while 21% and 27% showed moderate or weak staining, respectively [32]. Additionally, Zang et al. stained 803 prostate cancer samples and reported a median B7-H3 expression of 80%. Of these, 26% showed strong staining, 39% moderate staining, 27% weak staining, and 7% were negatively stained. Moreover, they found that patients with strong B7-H3 staining intensity were significantly more likely to develop seminal vesicle invasion, clinical recurrence, and lethal outcomes, compared to patients with weak staining [18]. These studies highlight the prognostic value of B7-H3 expression, as well as the crucial importance of selecting patients appropriately before administering B7-H3 therapies to enhance treatment efficacy.

A SPECT imaging radiotracer appears to be a useful tool to monitor B7-H3 expression, given its sensitive and non-invasive approach. ^99m^Tc, a validated and widely available short-lived radioisotope with a half-life of about 6 h and straightforward radiochemistry, is suitable and complements the rapid pharmacokinetics of sdAbs. Several preclinical studies have demonstrated that ^99m^Tc-radiolabeled sdAbs provide rapid SPECT imaging at 1 h after radiotracer injection, with good contrast and high tumor-to-background ratios [33,34,35]. SPECT imaging is extensively used in the clinic for nuclear imaging, due to its availability and cost-effectiveness. Nonetheless, positron emission tomography (PET) is increasingly considered a preferred clinical choice due to its high sensitivity, better contrast, and higher spatial resolution compared to SPECT [36].

This study describes the development and validation of anti-B7-H3 sdAbs as radiotracers for imaging human B7-H3^+^ tumors. The B7-H3 protein seems to be a challenging target to develop low molecular weight vectors that bind to it with high affinity. Li et al. screened eight large dromedary camelid libraries by phage display to isolate anti-B7-H3 sdAbs for CAR-T cell development. They identified only 3 sdAbs that were able to bind to native B7-H3 on cancer cells, though the affinities of the monovalent sdAbs were not reported [31]. Also, Oroujeni et al. designed a B7-H3 affibody for SPECT/CT imaging and recorded suboptimal binding affinities (EC_50_: 22 nM) and poor tumor retention at 4 h p.i., a time point typically ideal for imaging with affibody-based vectors [23]. We hypothesize that the challenge in generating sdAbs with high binding affinities from immune libraries could be associated with a close sequence similarity between camelids’ native B7-H3 and the human B7-H3 protein, leading to immune tolerance, limiting affinity maturation of the heavy-chain-only antibodies. We employed stringent second-generation panning conditions to select binders with improved affinity. This method is both time and cost-effective, delivering higher affinity binders without requiring extensive resources. This approach yielded the lead sdAb C51 with the highest binding affinity of 21.0 ± 0.7 nM. Although the applied strategy was effective, an alternative approach for selecting higher-affinity sdAbs includes introducing a competitor during the biopanning process [35]. Additionally, the affinity of the lead sdAb C51 can be further enhanced before clinical advancement using approaches such as site-directed CDR mutagenesis [37], alanine scanning [38], or computational affinity maturation [39] and phage display-based affinity maturation [24].

Binding ELISA confirmed cross-reactivity of all purified sdAbs to the 3 B7-H3 isoforms tested. Subsequent evaluation of the 4 lead compounds (Nb44, Nb55, C51, C80) for cross-reactivity on the murine and human B7-H3 isoforms by SPR, and flow cytometry on CT26 cells transduced to express m2IgB7-H3, corroborated the ELISA results. Nb0 exhibited a preference for the m2IgB7-H3 isoform and had poor or no binding to the h2IgB7-H3 isoform. Multi-specific sdAbs are vital in the context of non-invasive imaging of B7-H3 tumors. Although it is believed that the h4IgB7-H3 is the predominant human isoform, the expression of the h2IgB7-H3 isoform in certain cancers [6,40] highlights the importance of selecting a vector capable of recognizing both isoforms for accurate and sensitive diagnosis. The ability of our lead sdAb to recognize both human B7-H3 isoforms provides a significant diagnostic advantage, enabling reliable imaging of a broad spectrum of B7-H3-expressing tumors with a single tracer. Moreover, this isoform versatility not only streamlines diagnostic development by eliminating the need to engineer bi-specific constructs or separate vectors for each isoform, but it also enhances translational efficiency and clinical applicability across diverse tumor types where isoform expression may vary.

The lead sdAbs were radiolabeled with ^99m^Tc and evaluated for biodistribution and tumor targeting efficiency in mice bearing subcutaneous GBM tumors. 1 h p.i SPECT/CT imaging revealed specific accumulation of all [^99m^Tc]Tc-sdAb tracers in the tumor, with associated low background signals. Ex vivo biodistribution at 1.5 h p.i. showed tumor uptake between 1.5 h and 4.9%IA/g, with the [^99m^Tc]Tc-C51 demonstrating superior uptake compared to other tracers, including the [^99m^Tc]Tc-Nb0, with significant tumor uptake, high signal-to-blood and muscle ratios already at 1.5 h p.i., compared to the [^99m^Tc]Tc-R3B23. Despite low tracer uptake in normal tissues observed via SPECT/CT and ex vivo dissection studies, some toxicity may still occur when B7-H3-directed pharmaceuticals are used for therapy. Meeus et al. recently reported normal tissue toxicity in a GBM tumor model treated with B7-H3-targeted NanoCAR-T cells, even without detectable B7-H3 expression on SPECT/CT [13].

To establish the broad applicability of the lead tracer for B7-H3 detection, we conducted a proof-of-concept study using a melanoma subcutaneous tumor model. The lead tracer [^99m^Tc]Tc-C51 demonstrated specific accumulation and high imaging contrast compared to the irrelevant control [^99m^Tc]Tc-R3B23. Ex vivo dissection studies revealed significantly higher tumor uptake for the lead tracer (*p* = 0.0011) compared to the control. The difference in tumor uptake of [^99m^Tc]Tc-C51 in the melanoma tumor model, at 1.44 ± 0.11%IA/g, compared to 4.96 ± 1.4%IA/g in the GBM model, can be attributed to differences in in vivo B7-H3 expression.

We observed high radioactivity accumulation in the kidneys, which is typical of sdAb-based radiopharmaceuticals due to rapid renal clearance and reuptake mechanisms by the proximal tubular cells [34,41]. Moreover, the presence of the 6-His tags on sdAbs likely contributed to the elevated kidney uptake, as previously reported [42]. Owing to the small sizes of sdAbs (~15 kDa), which fall below the glomerular filtration threshold of ~65 kDa, they are freely filtered through the glomerular membrane, with the negatively charged megalin and cubulin receptors playing a role in reabsorbing the tracers into the proximal tubular cells, resulting in high radioactivity accumulation [43,44]. Different strategies have been explored and can be implemented to reduce the kidney retention observed with our tracers. These include increasing the circulatory half-life of sdAbs by conjugating them with specific linkers, like polyethylene glycol (PEG) [45], eliminating the 6-His tag, and blocking megalin/cubulin-mediated endocytosis by co-infusing the radiopharmaceutical with the positively charged amino acids (L-lysine or L-arginine) or the plasma expander Gelofusin^®^ [42,46]. Other strategies include the introduction of cleavable linkers between the sdAb and the radionuclide [47] and the implementation of the so-called pre-targeting approach [48].

To our knowledge, this is the first sdAb-based tracer developed for nuclear imaging of B7-H3-expressing tumors. However, a few mAbs- and affibody-based radiotracers have been preclinically evaluated for nuclear imaging of B7-H3^+^ tumors. These include [^89^Zr]Zr-DFO-hu4G4, [^89^Zr]Zr-DFO-Ab-1, [^89^Zr]Zr-DFO-Ab-2, [^99m^Tc]Tc-SYNT-179 and [^99m^Tc]Tc-AC12-GGGC [21,22,23,24]. These radiotracers recorded specific tumor accumulation in preclinical models, allowing PET and SPECT imaging of B7-H3 tumors. The long circulatory half-life of the mAb radiotracers resulted in an overall high tumor uptake. However, a downside of mAbs is the unfavorable delay in performing imaging with efficient contrast (usually > 48 h p.i). Also, the long circulatory half-life results in a high radiation burden to non-target organs, precipitated by using radionuclides with long half-life (Zirconium-89). Our lead SdAb tracer (C51) showed an overall lower tumor uptake compared to the mAb tracers; however, it provided specific tumor accumulation, allowing imaging with high contrast already at 1 h p.i., with low radioactivity burden to non-target organs/tissues except for the kidneys. Moreover, except for Ab-1 [21], reported to bind to the m2IgB7-H3, all the mAb compounds lack cross-binding to both human B7-H3 isoforms, limiting their broad applicability compared to the C51 sdAb.

Although the affibody tracers ([^99m^Tc]Tc-AC12-GGGC and [^99m^Tc]Tc-SYNT-179) demonstrated earlier imaging at 4 h p.i., compared to mAbs, our compound provided earlier time point imaging. Additionally, the cross-reactivity of the affibody molecules to both human B7-H3 isoforms were not reported, thus posing a similar limitation as mAb-based tracers in detecting cancers expressing both human isoforms. Moreover, while the lead affinity matured affibody compound (SYNT-179) demonstrated a better EC_50_ value on SK-OV-3 cells (7.5 nM) compared to our lead compound (C51: 48 nM), the tumor uptake of [^99m^Tc]Tc-C51 at 1.5 h p.i. (4.9 ± 1.41%IA/g) is comparable to the value obtained for [^99m^Tc]Tc-SYNT-179 (4.5 ± 0.56%IA/g) at 2 h p.i [24]. Importantly, [^99m^Tc]Tc-C51 provided imaging with high contrast at 1 h p.i., with low organ accumulation of radioactivity compared to 4 h p.i. for [^99m^Tc]Tc-SYNT-179.

In this study, we demonstrated that sdAbs, with small sizes and short biological half-life, paired with short-lived radioisotopes, minimized the radiation burden to non-target organs for non-invasive imaging applications. The lead sdAb (C51) tracer selectively accumulated in B7-H3^+^ tumors across two models, enabling early time point imaging with high signal-to-noise ratio and minimal off-target organ uptake, supporting its suitability for nuclear imaging. Importantly, the isoform versatility of the radiotracer demonstrates its broad applicability potential for cancers that express both human B7-H3 isoforms, though this remains to be evaluated in relevant tumor models. We employed SPECT imaging in this study; nonetheless, the sdAbs can also be designed into PET tracers by radiolabeling them with Gallium-68, Copper-64, or Fluorine-18 for PET imaging [41,49]. PET offers superior attenuation correction for image reconstruction, higher spatial resolution (5–7 mm vs. 10–14 mm for SPECT), and better sensitivity, detecting approximately three orders of magnitude more events within a given region than SPECT [36]. Due to the clinical relevance of PET, our team is actively developing the lead sdAb as a PET tracer to enhance its sensitivity for B7-H3 tumor detection. Additionally, to enhance B7-H3 detection sensitivity and minimize false negatives, this sdAb can be designed into a bimodal imaging tracer, combining complementary modalities for more accurate and reliable tumor visualization [50,51]. Furthermore, the versatile nature of these sdAbs supports their development into a range of targeted therapies, including ADCs (NCT05830123), nano-CAR-T cells (NCT04077866), and TRT (NCT03275402).

## 4. Materials and Methods

### 4.1. Cell Lines

All cells were cultured at 37 °C in a humidified atmosphere with 5% CO_2_. The GBM cell line U87-MG and the ovarian adenocarcinoma cell line SK-OV-3 were obtained from the American Type Culture Collection (ATCC, Manassas, VA, United state of America (USA), while the LN-229, 624MEL, and the m2IgB7-H3 transduced CT26 cells [13] were a kind donation from Karine Breckpot (VUB, Ixelles, Belgium). U87-MG and LN-229 cells and transduced CT26 cells were cultured in Dulbecco’s Modified Eagle Medium (DMEM: Thermofisher Scientific, Oxford, United kingdom (UK)), and SK-OV-3 in Roswell Park Memorial Institute medium (RPMI, Thermofisher Scientific, Oxford, UK). All media were supplemented with 10% fetal bovine serum (FBS, Haarlem The Netherlands), 2 mM L-glutamine (Sigma Aldrich, Overijse, Belgium), 100 U/mL penicillin, and 100 µg/mL Streptomycin (Sigma Aldrich, Belgium).

### 4.2. Single-Domain Antibody Generation, Identification, and Production

In collaboration with the nanobody service facility (NSF, VIB Belgium), two Ilamas were subcutaneously inoculated 6 times with recombinant extracellular domain of h4IgB7-H3 (Sino Biologicals, Beijing, China). After immunizations, peripheral blood was collected, and DNA coding for sdAb fragments was amplified for phage display library construction and biopanning [52]. The sdAbs were all produced with a C-terminal hexa-histidine tag (6-His) and purified using immobilized metal affinity chromatography (IMAC) followed by size exclusion chromatography. Details regarding sdAb generation, production, and purification are available in the Appendix A.

### 4.3. ELISA

A day before the experiment, 96-well plates (NUNC Maxisorp, ThermoFisher, Waltham, MA, USA) were coated with 1 µg/mL of the 3 B7-H3 recombinant proteins: h4IgB7-H3 (cat. 11188-H08H, Sino Biologicals, Beijing, China), h2IgB7-H3 (cat. 50973-M08H, Sino Biologicals, Beijing, China), and m2IgB7-H3 (cat. 1949-B3, R&D Systems, Minneapolis, MN, USA). The coated plates were incubated overnight at 4 °C. The plates were washed with PBS-T (PBS + 0.05% Tween 20) (Merck-Millipore, Burlington, MA, USA). Non-specific binding was blocked by incubating the plates with 2% skimmed milk at room temperature (RT) for 2 h (h), followed by adding Haemagglutinin (HA) tagged-sdAb extract or 2 µg of purified 6-His-tagged sdAb in a final volume of 100 µL PBS. Next, binding was detected using mouse 1:2000 anti-HA antibodies (clone 16B12, Biolegend, San Diego, CA, USA) or anti-HIS (Biolegend, San Diego, CA, USA) antibody, followed by adding a 1:2000 dilution of anti-mouse alkaline phosphatase-conjugated antibody (cat. A3562, Sigma Aldrich, Overijse, Belgium). One or more washing steps with PBS-T were performed between every incubation step. Binding was developed by adding 100 µL of 2 mg/mL phosphatase substrate (PNPP substrate, ThermoFisher, Oxford, UK) dissolved in AP-blot buffer to the wells. Absorbance was measured at 405 nM using the Varioskan^TM^ LUX ELSIA reader (ThermoFisher, Oxford, UK).

### 4.4. Flow Cytometry

To evaluate the expression of B7-H3 on U87-MG, LN-229, SK-OV-3, and 624MEL, cells were harvested and washed twice with FACS buffer (Phosphate-Buffered Saline supplemented with 2 mM Ethylenediaminetetraacetic Acid (EDTA) and 2% FBS). Next, 1 × 10^5^ cells per condition (unstained, stained, and isotype) were incubated at 4 °C for 30 min with 1:100 anti-human B7-H3 antibodies (clone MIH42, Biolegend, San Diego, CA, USA) and isotype control (clone MIH35, Biolegend, San Diego, CA USA). After incubation, the cells were washed twice with FACS buffer, and data were acquired using a FACS Celesta flow cytometer (BD Biosciences, San Jose, CA, USA) and analyzed with FlowJo version 10.6.2. (Tree Star Inc., Ashland, OR, USA).

To determine the affinity of sdAbs on cells, 10^5^ U87-MG and SK-OV-3 cells/well in 50 μL FACS buffer were added to a 96 U-bottom plate. Next, 50 μL of a 1/3 serial dilution (1.8–4000 nM) of sdAbs was added and incubated at 4 °C for 1 h. The cells were washed twice with FACS buffer, and binding was detected by adding a 1:2000 dilution of PE-labeled anti-His antibody (clone GG11-8F3.5.1, Miltenyl Biotec, Bergisch Gladbach, Germany), followed by 30 min of incubation at 4 °C. Next, the cells were washed and resuspended in 250 μL FACS buffer. Results were acquired using a high-throughput BD FACS Celesta flow cytometer (BD Biosciences) and analyzed with FlowJo version 10.6.2 (Tree Star Inc., Ashland, OR, USA). Results were plotted as percentage PE positive using GraphPad Prism software version 9.5.0.

### 4.5. Surface Plasmon Resonance (SPR)

The affinity and kinetics of the purified sdAbs on the 3 B7-H3 recombinant proteins were determined by SPR on a Biacore T200 instrument (Cytiva, Uppsala, Sweden). 15 μg/mL of recombinant proteins dissolved in 10 mM sodium acetate with pH 4.0 were immobilized via amine-coupling chemistry on a CM5 sensor chip (Cytiva, Uppsala, Sweden) at 400–800 response units. Binding studies were performed with 1:2 serial dilutions of the different sdAbs, ranging between 1.9 nM and 250 nM concentrations. HEPES buffer (0.15 M NaCl, 3 mM EDTA, 0.01 M HEPES, 0.005% Tween-20, pH 7.4) at a 30 µL per minute flow rate was used as running buffer. The sdAbs were injected sequentially without regeneration cycles, with binding and dissociation set at 100 s and 300 s, respectively. Curves were fitted with Biacore’s T200 evaluation software (Cytiva) at a 1:1 ratio (antigen:ligand) interaction model with drift and RI2 correction, which was used to determine association rate constants (k_a_), dissociation rate constants (k_d_), and equilibrium dissociation constants (K_D_).

### 4.6. Single-Cell Suspension of Tumors

Single-cell suspension to evaluate B7-H3 expression using flow cytometry in implanted U87-MG tumors was performed using the Miltenyi Biotec protocol. More details can be found in the Appendix A.

### 4.7. Thermal Stability Assay

0.2 mg/mL to 0.6 mg/mL anti-B7H3 sdAbs in triplicate were mixed with 5000X SYPRO^TM^ orange protein gel stain (ThermoFisher Scientific) diluted to 1× final concentration using PBS. The samples were added to a white 96-well PCR plate. Fluorescence signals were captured using the CFX connect PCR device (BioRad, Hercules, CA, USA), with a 0.5 °C per minute temperature increment starting from 25 °C to 95 °C.

### 4.8. Radiolabelling of sdAbs with Technetium-99m (^99m^Tc)

The sdAbs were labeled with ^99m^Tc using the tricarbonyl method as previously described [53]. Briefly, [^99m^Tc]TcO_4_^-^ eluate from a Molybdenum-99m generator was incubated with an Isolink kit (Paul Scherrer Institute, Villgen, Switzerland) at 100 °C for 20 min. Next, the ^99m^Tc-tricarbonyl was complexed to the C-terminal 6-His tag of the sdAb by incubating 50 µg of sdAb at 50 °C for 90 min. Purification of the ^99m^Tc-sdAb was performed by running the samples through a NAP-5 column (Cytiva), followed by filtration through a 0.22 µM pore filter (Millipore, Haren, Belgium).

The radiochemical purity (RCP) of the labeled sdAbs was evaluated by an instant thin-layer Chromatograph (iTLC) before and after purification. Briefly, 2 µL of ^99m^Tc-sdAbs were spotted on an iTLC strip (a 15 mm × 200 mm silica gel strip impregnated glass fiber sheet), followed by dipping the strip into 100% acetone solution, which acts as the mobile phase. Complexed ^99m^Tc-sdAbs remain at the bottom of the strip where the product is spotted, while uncomplexed ^99m^Tc migrates with the mobile phase to the top of the strip. The distribution of radioactivity over the strip was analyzed using the Elysia-Raytest miniGita 37,038 software 6.3. An RCP limit of >95% was used for experiments. Next, the radiochemical yield (RCY) was calculated by expressing the decay-corrected activity of the product as a percentage of the initial ^99m^Tc activity added at the start of the reaction.

### 4.9. Radioligand-Specific Cell Binding Assay

Two days before the experiment, 10^5^ cells/well were seeded in triplicate in 12-well plates and incubated at 37 °C in humidified air and 5% CO_2_. One hour before the experiment, cells were placed at 4 °C. Next, the medium was removed, and the cells were washed twice with ice-cold PBS. 10 µM of cold sdAb was added to blocked wells, followed by the addition of 100 nM ^99m^Tc-sdAb to both the blocked and unblocked wells. The cells were incubated at 4 °C for 1 h. Next, the supernatant containing unbound sdAbs was collected into labeled tubes, and the cells were washed twice with ice-cold PBS. Subsequently, cells were lysed by incubating with 1 M NaOH, and the lysates were collected. The lysis process was repeated, and the radioactivity in the collected fractions (bound and unbound) was counted using a gamma counter (Perkin-Elmer, Waltham, MA, USA).

### 4.10. Mouse Model

Female 6–12-week-old mice Nu(ncr)Foxn1nu were purchased from Charles River Laboratories (Ecully, France). For the biodistribution study in tumor-bearing mice, 2 × 10^6^ U87-MG cells were subcutaneously inoculated on the right flank in unsupplemented medium with 50% Matrigel (Corning, Bedford, MA, USA). Meanwhile, 3.5 × 10^6^ 624MEL cells were subcutaneously inoculated in PBS without Matrigel. The welfare of the animals was checked daily, and tumor growth was monitored by caliper measurements. The formula (Length × Width^2^)/2 was used to calculate the tumor volumes. All animal studies were approved by the ethical committee on animal experimentation of the Vrije Universiteit Brussels (ECD: 22-272-02).

### 4.11. Micro-SPECT-CT Imaging

Tumor-bearing mice were i.v. injected with ~5 µg of ^99m^Tc-sdAb (54 ± 12.7 MBq). 1 h p.i., imaging was performed by initially sedating the mice with 75 mg/kg ketamine and 1 mg/kg medetomidine (Ketamidor, Wels, Austria). The Vector^+^ scanner (MILABS, Houten, The Netherlands) was used for micro-SPECT/CT imaging. To do this, the 1.5 mm 75-pinhole collimator, with 6 bed positions, was selected. SPECT imaging was performed with a 150 s per bed position for a total of 15 min, followed by CT acquisition conducted at 60 kV and 615 mA for 2 min. Images were reconstructed using MILABS’ reconstruction software V8.00, using a 0.4 voxel size, 2 subsets, and 4 iterations. Visual analysis was performed using AMIDE software version 1.0.5 (UCLA, Los Angeles, CA, USA) and VivoQuant software version 5.4.3 (Invicro, Needham, MA, USA). The mice were killed at 1.5 h p.i. for ex vivo biodistribution studies. To this end, the tumor and various organs/tissues were harvested, and the radioactivity in each organ was counted using a gamma counter (Perkin ELMER, Waltham, MA, USA). Results were expressed as percentage injected activity per gram organ/tissue (%IA/g).

### 4.12. Statistical Analysis

Student *t*-tests, non-linear regression analysis, and one-way ANOVA with Dunnett’s test were used to test for statistical significance on GraphPad Prism software. Results were expressed as mean ± standard deviation, with *p*-values *p* < 0.05 (*), *p* < 0.01 (**), *p* < 0.001 (***), and *p* < 0.0001 (****) considered significant. The *p*-values were adjusted to account for multiple comparisons to avoid Type I errors.

## Figures and Tables

**Figure 1 pharmaceuticals-18-01477-f001:**
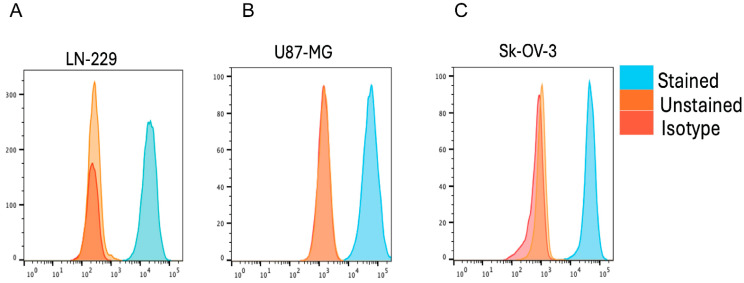
Flow cytometry analysis confirmed endogenous human B7-H3 expression on (**A**) LN-229, (**B**) U87-MG, and (**C**) SK-OV-3 cells.

**Figure 2 pharmaceuticals-18-01477-f002:**
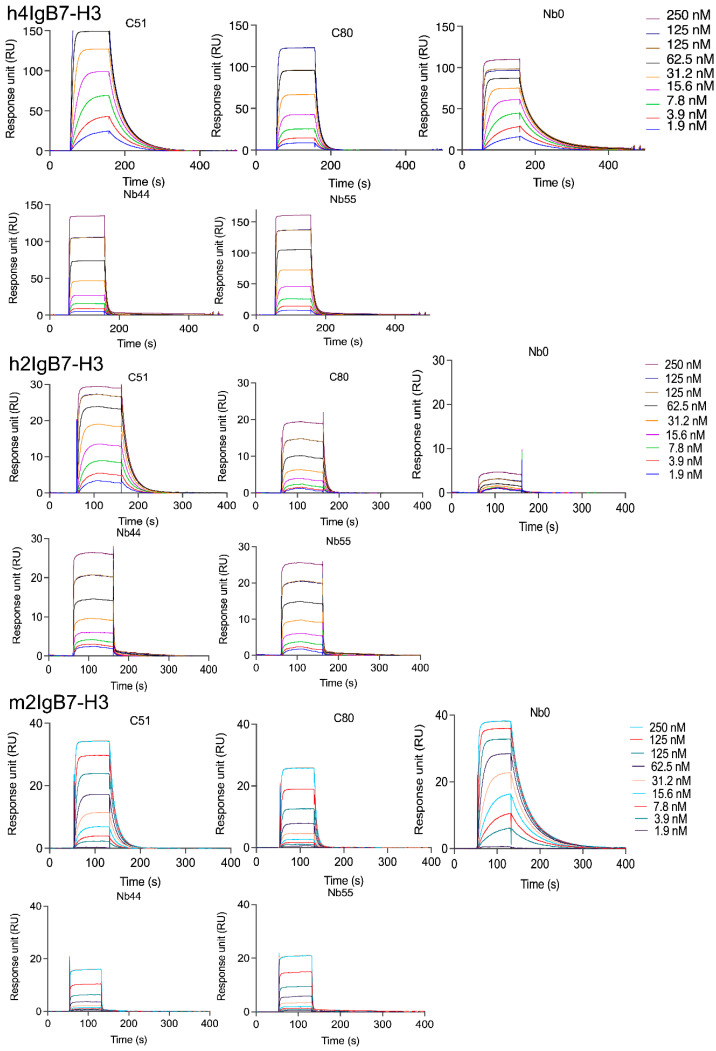
Sensorgrams (n = 3) of lead sdAbs dynamically interacting with the 3 tested B7-H3 recombinant proteins (h4IgB7-H3, h2IgB7-H3, m2IgB7-H3). All sdAbs are cross-reactive to all isoforms, except for Nb0, which does not bind to h2IgB7-H3.

**Figure 3 pharmaceuticals-18-01477-f003:**
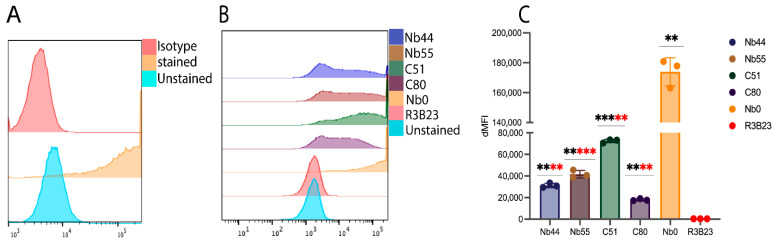
Binding of lead sdAbs to m2IgB7-H3 on transduced CT26 cell lines: (**A**) Flow cytometry analysis using commercially available mouse anti-B7-H3 antibody on transduced CT26 cells confirmed expression of m2IgB7-H3. (**B**) sdAbs showed specific targeting of m2IgB7-H3 on CT26 transduced cells (n = 3). (**C**) ΔMFI comparison between sdAbs. Black asterisks (*) indicate a significant difference between ΔMFI of sdAbs compared to R3B23, and red asterisks (*) indicate a significant difference between ΔMFI of sdAbs compared to Nb0. **, *p*-value < 0.01; ***, *p*-value < 0.001.

**Figure 4 pharmaceuticals-18-01477-f004:**
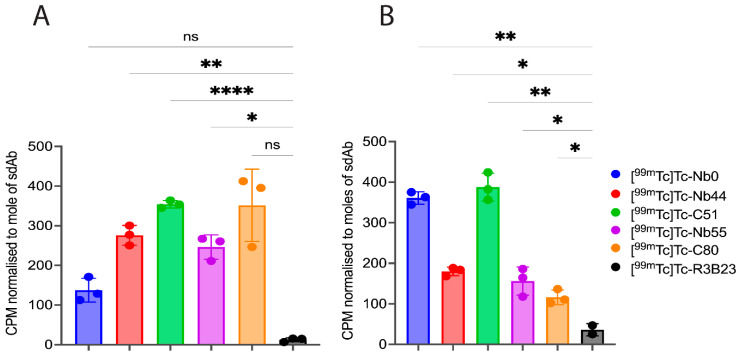
Specific binding of ^99m^Tc-labeled sdAbs on (**A**) U87-MG and (**B**) SK-OV-3 cells (n = 3). Radioactivity counts (CPM) are normalized to molar specific activity for each compound and presented as mean ± SD. ns, *p*-value > 0.05; *, *p*-value < 0.05; **, *p*-value < 0.01; ****, *p*-value < 0.0001.

**Figure 5 pharmaceuticals-18-01477-f005:**
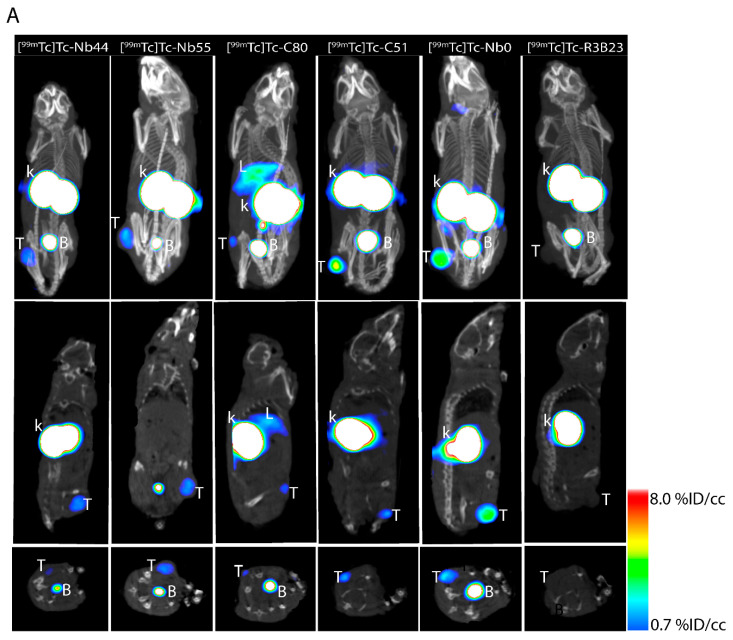
In vivo tumor targeting and biodistribution in U87-MG subcutaneous tumor model (n = 5): (**A**) micro-SPECT/CT imaging at 1 h p.i. Results revealed specific accumulation in the tumors and high signals from the bladder and kidneys (signal threshold set at 0.7–8% ID/cc). Images are presented as MIP, Sagittal, and transverse views, from top to bottom. The white blot on the kidneys and bladder is due to the high signals above the threshold defined in the figure legend. (**B**) Tumor uptake comparison across compounds. (**C**) Ex vivo biodistribution in normal organs/tissues. Results are obtained by quantifying the activity in each organ using a gamma counter and expressing it as percentage injected activity per gram (%IA/g) (n = 5). T, tumor; k, kidneys; B, bladder; L, liver. ns, *p*-value > 0.05; *, *p*-value < 0.05; **, *p*-value < 0.01.

**Figure 6 pharmaceuticals-18-01477-f006:**
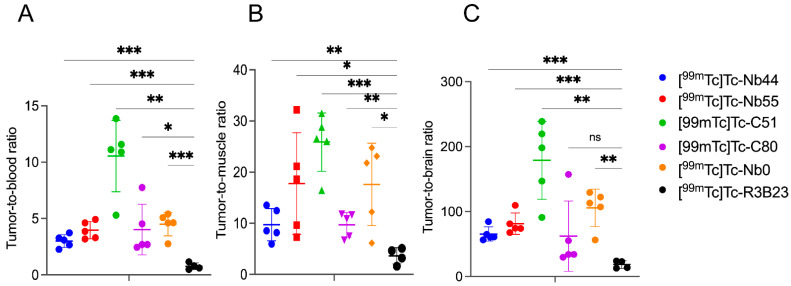
Tumor-to-organ ratios of ^99m^Tc-labeled sdAbs: (**A**) Tumor-to-blood (**B**) Tumor-to-muscle (**C**) Tumor-to-brain ratios. Results are obtained by dividing the %IA/g of the tumor by each organ (n = 5). ns, *p*-value > 0.05; *, *p*-value < 0.05; **, *p*-value < 0.01; ***, *p*-value < 0.001.

**Figure 7 pharmaceuticals-18-01477-f007:**
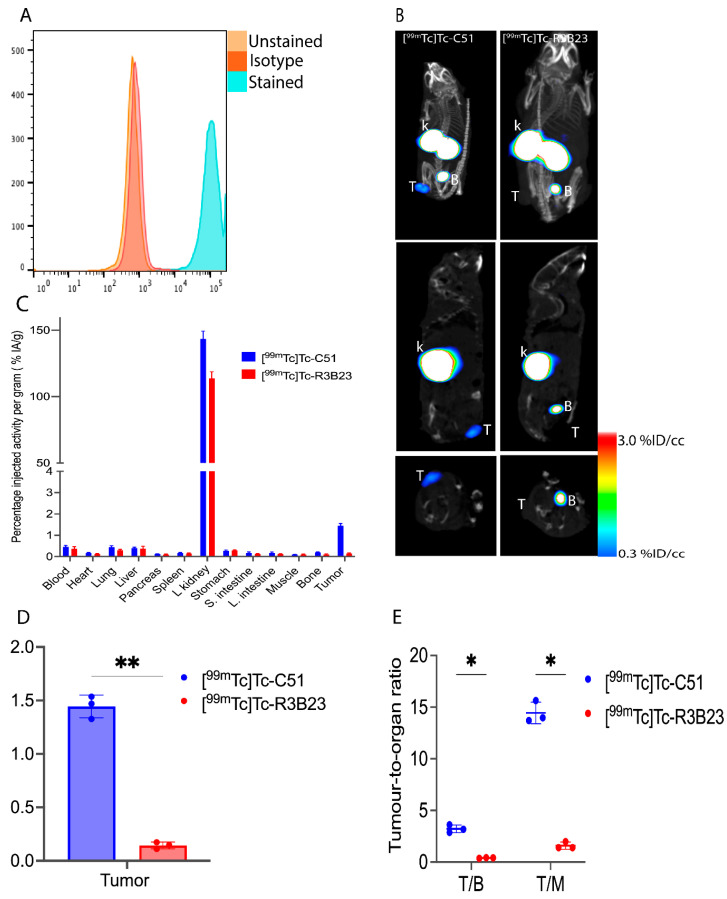
In vivo tumor targeting and biodistribution in 624MEL subcutaneous tumor model (n = 3): (**A**) Flow cytometry confirmed 624MEL cell line expressed native B7-H3 protein, (**B**) micro-SPECT/CT imaging at 1 h p.i. Results revealed specific accumulation in the tumors and high signals from the bladder and kidneys (signal threshold set at 0.3–3%ID/cc). Images are presented as MIP, Sagittal, and transverse views, from top to bottom. The white blot on the kidneys and bladder is due to the high signals above the threshold defined in the figure legend. (**C**) Ex vivo biodistribution in normal organs/tissues. Results are obtained by quantifying the activity in each organ using a gamma counter and expressing them as percentage injected activity per gram (%IA/g), (n = 3). (**D**) Tumor uptake comparison showed significantly higher tumor uptake for [^99m^Tc]Tc-C51 compared to [^99m^Tc]Tc-R3B23. (**E**) Comparison of tumor-to-blood and tumor-to-muscle ratios (n = 3). T, tumor; k, kidneys; B, bladder. ns, *p*-value > 0.05; *, *p*-value < 0.05; **, *p*-value < 0.01.

**Table 1 pharmaceuticals-18-01477-t001:** Biophysical and affinity properties of first- and second-generation sdAbs: Affinity on recombinant h4IgB7-H3 proteins as determined by SPR (n = 1), K_D_ values on U87-MG and SK-OV-3 cells using flow cytometry (n = 1) and melting temperatures (n = 3). Gray highlighted sdAbs are the 4 lead compounds selected for further characterization.

SdAb	Affinity and Kinetics Parameters on 4IgB7-H3			K_D_ Values on Cell (nM)		Melting Temperature (°C)
	k_a_ (1/M.s)	k_d_ (1/s)	K_D_ (nM)	U87-MG	SK-OV-3	
First-generation sdAbs						
Nb2	2.08 × 10^6^	1.34 × 10^−1^	341	121	33	61.0 ± 0.2
Nb16	2.08 × 10^6^	1.34 × 10^−1^	126	178	228	71.0 ± 0.5
Nb29	1.84 × 10^5^	2.40 × 10^−2^	130	115	23	70.0 ± 0.2
Nb38	2.49 × 10^6^	2.31 × 10^−1^	92	325	257	65.0 ± 0.3
Nb39	1.28 × 10^6^	1.52 × 10^−1^	118	114	108	56.0 ± 0.3
Nb43	9.26 × 10^5^	1.07 × 10^−1^	115	256	66	54.0 ± 00
Nb44	2.23 × 10^6^	2.29 × 10^−1^	102	244	79	77.0 ± 00
Nb55	2.90 × 10^6^	1.96 × 10^−1^	67	289	22	66.0 ± 0.1
Nb62	9.85 × 10^5^	9.87 × 10^−2^	100	289	29	53.0 ± 0.3
Nb67	9.72 × 10^5^	1.08 × 10^−1^	111	240	75	54.0 ± 0.5
Nb68	6.83 × 10^5^	1.11 × 10^−1^	161	219	158	52.0 ± 0.3
Nb75	1.86 × 10^6^	2.60 × 10^−1^	140	895	731	78.0 ± 0.2
Nb78	2.16 × 10^6^	2.28 × 10^−1^	105	104	66	71.0 ± 0.2
Second-generation sdAbs						
C22	1.30 × 10^6^	1.16 × 10^−1^	89	198	265	55.0 ± 0.3
C51	2.39 × 10^6^	5.78 × 10^−2^	24	20	48	55.0 ± 0.4
C80	1.42 × 10^6^	1.14 × 10^−1^	80	159	363	54.8 ± 0.2
Benchmark sdAb						
Nb0	2.37 × 10^6^	3.36 × 10^−2^	14	36	42	68.0 ± 00

## Data Availability

The datasets generated and analyzed for this study are available online at https://doi.org/10.5281/zenodo.17220286, and can be provided upon reasonable request from the corresponding authors.

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
