# Peer review of "Design and Preclinical Validation of an Anti-B7-H3-Specific Radiotracer: A Non-Invasive Imaging Tool to Guide B7-H3-Targeted Therapies"

_pharmaceuticals, 2025, doi:10.3390/ph18101477_

Round 1
Reviewer 1 Report
Comments and Suggestions for Authors
Linked to anti-cancer immunity and pro-tumoral activity and usually corresponding with negative patient prognosis, B7-H3 is an immunoregulatory protein belonging to the B7 family. Although often expressed in several cancer types, B7-H3 shows both inter- and intra-tumoral expression heterogeneity, which could lower the therapeutic efficacy of treatments aiming at B7-H3. Consequently, a clear need exists for a sensitive, non-invasive method to evaluate B7-H3 expression to guide the choice and stratification of people who might profit from anti-B7-H3 treatment. Perfect vectors for the development of a diagnostic tool are single-domain antibody fragments (sdAbs). The development and thorough preclinical validation of a new Technetium-99m-labeled single-domain antibody (sdAb) tracer targeting B7-H3 (CD276), a checkpoint protein routinely overexpressed in a wide spectrum of malignancies, is presented in this work. Using micro-SPECT/CT in a glioblastoma (U87-MG) mouse model, the authors effectively show that the lead candidate sdAb, C51, offers isoform-cross-reactive, fast, and specific tumor imaging capability. The work fills in a clear diagnostic void in the patient stratification for B7-H3-targeted therapies.
The authors address a major clinical challenge—heterogeneous B7-H3 expression—using a logical diagnostic tool based on sdAbs. Two generations of phage display, ELISA, SPR, flow cytometry, radiolabeling, and in vivo SPECT imaging are used in the study using appropriate and well-applied experimental approaches. Different from earlier studies using either complete antibodies or affibody molecules, this is the first known sdAb-based SPECT tracer for B7-H3 imaging. Combining the fast elimination of sdAbs with [99mTc], a therapeutically accessible, short-lived radioisotope, helps to enable immediate practical application. A major feature for preclinical and clinical adaptability is that the main sdAb (C51) binds to both main human B7-H3 isoforms and their murine counterparts.
The manuscript has several defects: Incorporate or examine findings from at least one more B7-H3+ tumor model (e.g., breast or prostate cancer) to demonstrate more general relevance. Though the use of 99mTc is justified, a comparison of PET tracers would greatly enhance the diagnosis. Although characteristic of sdAbs, propose ways to alleviate renal retention (e.g., His-tag elimination, co-administration of gelofusine), particularly if treatment modification is anticipated. Propose feasible affinity maturation or engineering strategies to augment binding strength for forthcoming clinical advancement. Advice on professional editing for grammatical accuracy, tense uniformity, and abstract brevity. The discourse can be condensed and become more interpretive. Figures must be enhanced to publication quality (≥300 dpi); SPECT scans should distinctly delineate tumor areas. A graphical abstract or workflow diagram is recommended. The manuscript's English proficiency is generally scientifically sufficient, although it does not meet the highest academic writing requirements in certain aspects. It would greatly benefit from expert language refinement to enhance readability, coherence, and accuracy.
Comments on the Quality of English LanguageThe manuscript's English proficiency is generally scientifically sufficient, although it does not meet the highest academic writing requirements in certain aspects. It would greatly benefit from expert language refinement to enhance readability, coherence, and accuracy.
Author Response
Response to Reviewer 1 Comments
- Summary
Thank you very much for taking the time to review this manuscript. Please find the detailed responses below and the corresponding revisions/corrections highlighted/in track changes in the re-submitted files.
- Point-by-point response to Comments and Suggestions for Authors
- Comments 1: Incorporate or examine findings from at least one more B7-H3+ tumor model (e.g., breast or prostate cancer) to demonstrate more general relevance.
Response 1: Thank you for pointing this out. Agree. We acknowledge that validating our lead tracer in a second tumor model, as well as in relevant models expressing the h2IgB7-H3 isoform, would enhance the overall relevance of the radiotracer. However, it is not possible to provide this data at this stage of the work. Accordingly, we have included this as a limitation of our study in the discussion section and made recommendations for future studies.
Discussion: (lines 422-431)
It should be noted that, despite the promising results presented in this study, some limitations warrant consideration. While the lead sdAb demonstrated broad reactivity towards all tested B7-H3 isoforms, it’s in vivo tumor targeting was validated in a single tumor model. To strengthen the translational relevance of these findings, evaluation in additional B7-H3 expressing tumor models is recommended. Also, although the sdAb exhibited cross-reactivity to multiple B7-H3 isoforms in vitro, its tumor targeting potential was not assessed in tumor models expressing the h2IgB7-H3 or the m2IgB7-H3 counterpart. Future studies should include these models to confirm the tracer’s versatility and to better understand its diagnostic potential across a broader range of tumor types and isoform profiles.
- Comments 2: Though the use of 99mTc is justified, a comparison of PET tracers would greatly enhance the diagnosis.
Response 2: Agree. We have added a comparison of SPECT versus PET in the discussion section, highlighting the clinical relevance of PET over SPECT, and the future direction of the lead sdAb, C51.
Discussion: (lines 409-418)
We employed SPECT imaging in this study; nonetheless, the sdAbs can also be designed into PET tracers by radiolabeling them with Gallium-68, Copper-64, or Fluorine-18 for PET imaging (41,49). PET offers superior attenuation correction for image reconstruction, higher spatial resolution (5-7mm vs 10-14mm for SPECT), and better sensitivity, detecting approximately three orders of magnitude more events within a given region than SPECT (36). Due to the clinical relevance of PET, our team is actively developing the lead sdAb as a PET tracer to enhance its sensitivity for B7-H3 tumor detection. To enhance B7-H3 detection sensitivity and minimize false negatives, this sdAb can be designed into a bimodal imaging tracer, combining complementary modalities for more accurate and reliable tumor visualization (50,51).
- Comments 3: Although characteristic of sdAbs, propose ways to alleviate renal retention (e.g., His-tag elimination, co-administration of gelofusine), particularly if treatment modification is anticipated.
Response 3: Agree. We have proposed mitigation strategies in the discussion that can be employed to reduce the accumulation of radioactivity in the kidneys to enhance the safety profiles of the radiotracers.
Discussion: (lines 364-375)
Owing to the small sizes of sdAbs (~15 kDa), which fall below the glomerular filtration threshold of ~65 kDa, they are freely filtered through the glomerular membrane, with the negatively charged megalin and cubulin receptors playing a role in reabsorbing the tracers into the proximal tubular cells, resulting in high radioactivity accumulation (43,44). Different strategies have been explored and can be implemented to reduce the kidney retention observed with our tracers. These include increasing the circulatory half-life of sdAbs by conjugating them with specific linkers, like polyethylene glycol (PEG) (45), eliminating the 6-His tag, and blocking megalin/cubulin-mediated endocytosis by co-infusing the radiopharmaceutical with the positively charged amino acid (L-lysine or L-arginine) or the plasma expander GelofusinÒ (42,46). Other strategies include the introduction of cleavable linkers between the sdAb and the radionuclide (47) and the implementation of the so-called pre-targeting approach (48).
- Comments 4: Propose feasible affinity maturation or engineering strategies to augment binding strength for forthcoming clinical advancement.
Response 4: Indeed, we have highlighted possible techniques that can be used to optimize the affinity of the lead sdAb for future applications.
Discussion : (lines 326-332)
Although the applied strategy was effective, an alternative approach for selecting higher affinity sdAbs involves introducing a competitor during the biopanning process [33]. The affinity of the lead sdAb C51 may be further enhanced before clinical advancement using approaches such as site-directed CDR mutagenesis (38), alanine scanning (39), computational affinity maturation (40) and phage display-based affinity maturation (24).
- Comments 5: Advice on professional editing for grammatical accuracy, tense uniformity, and abstract brevity.
Response 5: We have thoroughly revised and proofread the manuscript to improve grammatical accuracy, ensure consistent tense usage, and enhance the brevity and clarity of the abstract, as advised.
- Comments 6: The discourse can be condensed and become more interpretive.
Response 6: Thank you for this valuable feedback. We have made a concerted effort to condense the discussion section and make it more interpretive. Even though incorporating the reviewers’ suggestions did require the addition of some content, we have carefully streamlined the text to maintain focus and avoid unnecessary repetition.
Discussion:
- lines 313-321: This paragraph has been condensed.
- Paragraph 7 of the original document (lines 367-379) has been deleted, summarized into 3 lines, and added to the results section (lines 227-230).
- Paragraph 11, lines 424-431 of the original document have been condensed into two lines and added to paragraph 10, lines 419-420 of the current the manuscript
- Comments 7: Figures must be enhanced to publication quality (≥300 dpi); SPECT scans should distinctly delineate tumor areas.
Response 7: Agree. We have enhanced the quality of all the figures to the journal’s requirements.
- Comments 8: A graphical abstract or workflow diagram is recommended.
Response 8: A graphical abstract was uploaded to the submission platform at the time of submission. The journal does not advise adding the graphical abstract to the manuscript.
- Comments 9: The manuscript's English proficiency is generally scientifically sufficient, although it does not meet the highest academic writing requirements in certain aspects. It would greatly benefit from expert language refinement to enhance readability, coherence, and accuracy.
Response 9: As mentioned in point 5 above, the manuscript has been thoroughly revised and carefully proofread to improve tense consistency, clarity, readability, and overall coherence.

Reviewer 2 Report
Comments and Suggestions for Authors
1- Strengthen the Introduction and Discussion to clearly state what differentiates this study regarding isoform promiscuity, early imaging potential (1-hour p.i.), and cross-reactivity across human/mouse isoforms. Explicit comparison with previous mAb and affibody tracers (e.g., [99mTc]Tc-SYNT-179) would help contextualize the innovation.
2-Although, additional in vivo models may not be feasible at this stage, the Discussion should transparently acknowledge this limitation and suggest how future studies will address isoform-specific tumor validation in other cancer types. Including supportive in vitro data is helpful, but in vivo heterogeneity remains a challenge for translation.
3- In details, enhance the Discussion on renal clearance and potential strategies to reduce kidney retention (e.g., PEGylation, use of cleavable linkers, co-injection of Gelofusine or lysine). and also Addressing the limitations of SPECT sensitivity and proposing how PET tracers or dual-imaging approaches may mitigate risks related to false-negative expression detection.
4- enhance figure resolution and readability by increasing font size, simplifying axis labels, Consider relocating some details (e.g., melting curves, detailed ELISA traces) to Supplementary Figures. Highlight key comparisons (e.g., C51 vs. Nb0) clearly in the main figures or accompanying legends.
5-biliograph section is amazing some reference is not complete data such as (reference 34), also Add specific references when citing past studies with zirconium-labelled tracers or affibodies, especially when making comparative statements.
6-Clarify in the Conflicts of Interest or Funding section
Author Response
Response to Reviewer 2 Comments
- Summary
Thank you very much for taking the time to review this manuscript. Please find the detailed responses below and the corresponding revisions/corrections highlighted/in track changes in the re-submitted files.
- Point-by-point response to Comments and Suggestions for Authors
- Comments 1: Strengthen the Introduction and Discussion to clearly state what differentiates this study regarding isoform promiscuity, early imaging potential (1-hour p.i.), and cross-reactivity across human/mouse isoforms. Explicit comparison with previous mAb and affibody tracers (e.g., [99mTc]Tc-SYNT-179) would help contextualize the innovation.
Response 1: Thank you very much for this valuable insight. We have emphasized the importance of tracer promiscuity in the introduction and strengthened the comparison on the performance of our lead compound against reported mAbs and affibodies, in terms of early time point imaging, safety, and isoform promiscuity.
Introduction: (lines 93-98)
The ability of the aforementioned tracers to recognize both human B7-H3 isoforms was not reported, which may limit their sensitivity and clinical applicability for detecting tumors that express the h2IgB7-H3 isoform. Therefore, there remains an unmet need for a time-efficient, non-invasive nuclear imaging strategy capable of detecting tumors expressing both human B7-H3 isoforms.
Discussion: (lines 386-397)
Our lead SdAb tracer (C51) showed an overall lower tumour uptake compared to the mAb tracers; however, it provided specific tumour accumulation, allowing imaging with high contrast already at 1-hour p.i., with low radioactivity burden to non-target organs/tissues except for the kidneys. Moreover, except for Ab-1 (21), reported to bind to the m2IgB7-H3, all the mAb compounds lack cross-binding to both human B7-H3 isoforms, limiting their broad applicability when compared to our lead C51 sdAb. Although the affibody tracers ([99mTc]Tc-AC12-GGGC and [99mTc]Tc-SYNT-179) demonstrated earlier imaging at 4 hours p.i. Compared to mAbs, our compound provided earlier time point imaging. Additionally, the cross-reactivity of the affibody molecules to both human B7-H3 isoforms was not reported, thus posing a similar limitation as mAb-based compounds in detecting cancers expressing both human isoforms.
- Comments 2: Although additional in vivo models may not be feasible at this stage, the Discussion should transparently acknowledge this limitation and suggest how future studies will address isoform-specific tumor validation in other cancer types. Including supportive in vitro data is helpful, but in vivo heterogeneity remains a challenge for translation.
Response 2: Agree. We acknowledge that validating our lead tracer in a second tumor model, as well as in relevant models expressing the h2IgB7-H3 isoform, would enhance the overall relevance of the radiotracer. Accordingly, we have included this as a limitation of our study in the discussion section and made recommendations for future studies.
Discussion: (lines 425-434)
It should be noted that, despite the promising results presented in this study, some limitations warrant consideration. While the lead sdAb demonstrated broad reactivity towards all tested B7-H3 isoforms, it’s in vivo tumor targeting was validated in a single tumor model. To strengthen the translational relevance of these findings, evaluation in additional B7-H3 expressing tumor models is recommended. Also, although the sdAb exhibited cross-reactivity to multiple B7-H3 isoforms in vitro, its tumor targeting potential was not assessed in tumor models expressing the h2IgB7-H3 or the m2IgB7-H3 counterpart. Future studies should include these models to confirm the tracer’s versatility and to better understand its diagnostic potential across a broader range of tumor types and isoform profiles.
- Comments 3: In details, enhance the Discussion on renal clearance and potential strategies to reduce kidney retention (e.g., PEGylation, use of cleavable linkers, co-injection of Gelofusine or lysine). and also Addressing the limitations of SPECT sensitivity and proposing how PET tracers or dual-imaging approaches may mitigate risks related to false-negative expression detection.
Response 3: Agree. We have provided a detailed explanation of the renal retention observed with our radiotracers in the discussion section and have proposed mitigation strategies, as advised. Additionally, we have discussed the advantages of developing a PET tracer over a SPECT tracer, especially in the context of clinical advancement.
Discussion: (lines 364-375)
Owing to the small sizes of sdAbs (~15 kDa), which fall below the glomerular filtration threshold of ~65 kDa, they are freely filtered through the glomerular membrane, with the negatively charged megalin and cubulin receptors playing a role in reabsorbing the tracers into the proximal tubular cells, resulting in high radioactivity accumulation (43,44). Different strategies have been explored and can be implemented to reduce the kidney retention observed with our tracers. These include increasing the circulatory half-life of sdAbs by conjugating them with specific linkers, like polyethylene glycol (PEG) (45), eliminating the 6-His tag, and blocking megalin/cubulin-mediated endocytosis by co-infusing the radiopharmaceutical with the positively charged amino acid (L-lysine or L-arginine) or the plasma expander GelofusinÒ (42,46). Other strategies include the introduction of cleavable linkers between the sdAb and the radionuclide (47) and the implementation of the so-called pre-targeting approach (48).
Lines (410-420):
We employed SPECT imaging in this study; nonetheless, the sdAbs can also be designed into PET tracers by radiolabeling them with Gallium-68, Copper-64, or Fluorine-18 for PET imaging (46,47). PET offers superior attenuation correction for image reconstruction, higher spatial resolution (5-7mm vs 10-14mm for SPECT), and better sensitivity, detecting approximately three orders of magnitude more events within a given region than SPECT (36). Accordingly, our team is actively developing the lead sdAb as a PET tracer to enhance its sensitivity for B7-H3 tumor detection. To enhance B7-H3 detection sensitivity and minimize false negatives, this sdAb can be designed into a bimodal imaging tracer, combining complementary modalities for more accurate and reliable tumor visualization (48,49).
- Comments 4: Enhance figure resolution and readability by increasing font size, simplifying axis labels, Consider relocating some details (e.g., melting curves, detailed ELISA traces) to Supplementary Figures. Highlight key comparisons (e.g., C51 vs. Nb0) clearly in the main figures or accompanying legends.
Response 4: Agree. The quality of all images has been enhanced to meet the journal’s requirements. Also, ELISA and melting curve plots have been moved to the supplementary sheet for clarity.
- Comment 5: Bibliography section is amazing some reference is not complete data such as (reference 34), also Add specific references when citing past studies with zirconium-labelled tracers or affibodies, especially when making comparative statements.
Response 5: The references have been carefully reviewed and edited as advised.
- Comments 6: Clarify in the Conflicts of Interest or Funding section
Response 6: More context on the conflict of interest has been provided.
Line (590-592)
These companies leverage sdAbs for targeted radionuclide therapy and nuclear imaging.
